

# Phylogenomic relationship and evolutionary insights of sweet potato viruses from the western highlands of Kenya

James M. Wainaina[1], Elijah Ateka[2], Timothy Makori[2], Monica A. Kehoe[3] and Laura M. Boykin[1]

[1] School of Molecular Sciences/ARC CoE Plant Energy Biology, The University of Western Australia, Crawley, WA, Australia
[2] Department of Horticulture, Jomo Kenyatta University of Agriculture and Technology, Nairobi, Kenya
[3] Plant Pathology, Department of Primary Industries and Regional Development Diagnostic Laboratory Service, South Perth, WA, Australia

## ABSTRACT

Sweet potato is a major food security crop within sub-Saharan Africa where 90% of Africa production occurs. One of the major limitations of sweet potato production are viral infections. In this study, we used a combination of whole genome sequences from a field isolate obtained from Kenya and those available in GenBank. Sequences of four sweet potato viruses: *Sweet potato feathery mottle virus* (*SPFMV*), *Sweet potato virus C* (*SPVC*), *Sweet potato chlorotic stunt viru*s (*SPCSV*), *Sweet potato chlorotic fleck virus* (*SPCFV*) were obtained from the Kenyan sample. SPFMV sequences both from this study and from GenBank were found to be recombinant. Recombination breakpoints were found within the Nla-Pro, coat protein and P1 genes. The SPCSV, SPVC, and SPCFV viruses from this study were non-recombinant. Bayesian phylogenomic relationships across whole genome trees showed variation in the number of well-supported clades; within SPCSV (RNA1 and RNA2) and SPFMV two well-supported clades (I and II) were resolved. The SPCFV tree resolved three well-supported clades (I–III) while four well-supported clades were resolved in SPVC (I–IV). Similar clades were resolved within the coalescent species trees. However, there were disagreements between the clades resolved in the gene trees compared to those from the whole genome tree and coalescent species trees. However the coat protein gene tree of SPCSV and SPCFV resolved similar clades to the genome and coalescent species tree while this was not the case in SPFMV and SPVC. In addition, we report variation in selective pressure within sites of individual genes across all four viruses; overall all viruses were under purifying selection. We report the first complete genomes of SPFMV, SPVC, SPCFV, and a partial SPCSV from Kenya as a mixed infection in one sample. Our findings provide a snap shot on the evolutionary relationship of sweet potato viruses (SPFMV, SPVC, SPCFV, and SPCSV) from Kenya as well as assessing whether selection pressure has an effect on their evolution.

Corresponding author
Laura M. Boykin,
laura.boykin@uwa.edu.au

## INTRODUCTION

Sweet potato is grown in over nine million hectares (*Food and Agriculture Organization of the United Nations (FAO), 2016*) with 97% of global production confined to China and Africa (*FAOSTAT, 2006*). In Africa, 90% of the production occurs around the Lake Victoria region and in the western highlands of Kenya (*Ewell, 1960*; *Loebenstein, 2010*). Sweet potato is considered to be a food security crop and is grown within smallholder agro-ecosystems. It is intercropped with legumes such as beans (*Phaseolus vulgaris*), cowpea (*Vigna unguiculata*) and groundnut (*Arachis hypogaea L.*) particularly within smallholder farms in Africa. However, there is a two-fold difference in production levels between smallholder farms in Africa compared to Asia, and America (*Loebenstein, 2010*). One major reason for these differences is the spread of viral diseases within the cropping system. There are two primary modes of viral transmission within sweet potato. Sweet potato is vegetatively propagated, and through this there is the possibility of spreading viruses from the parent stock. The second mode of transmission is through viruliferous aphids in particular: *Aphis gossypii*, *Myzus persicae*, *Aphis craccivora* and *Lipaphis erysimi* and some whiteflies (*Bemisia tabaci*, *Trialeurodes vaporariorum*) (*Tugume, Mukasa & Valkonen, 2008*; *Navas-Castillo, Lopez-Moya & Aranda, 2014*).

Some of the major viruses affecting sweet potato production include: *Sweet potato feathery mottle virus* (SPFMV), genus *Potyvirus*, family *Potyviridae*, *Sweet potato chlorotic stunt viru*s (SPCSV), genus *Crinivirus*, family *Closteroviridae*, *Sweet potato mild mottle virus* (SPMMV), genus *Ipomovirus*, family *Potyviridae*, *Sweet potato virus C* (SPVC), genus *potyvirus*, family *Potyviridae*, and *Sweet potato chlorotic fleck virus* (SPCFV) genus *Carlavirus*, family *Flexiviridae* (*Tairo et al., 2005*). Of interest within the family *Potyviridae* and previously reported in the western highland of Kenya are SPFMV and SPVC, which are typical members of the genus *Potyvirus*. They are flexuous, non-enveloped, rod-shaped particles that are 680–900 nm long and 11–15 nm wide (*Urcuqui-inchima, Haenni & Bernardi, 2001*). They contain a single open reading frame (ORF) that is approximately 10,000 nucleotides (nt) and transcribes 10 genes with varying functions (*Urcuqui-inchima, Haenni & Bernardi, 2001*; *Wainaina et al., 2018*). On the other hand, SPCSV genus *Crinivirus*, family *Closteroviridae* has a non-enveloped bipartite genome (*Karasev, 2000*). The genome is composed of a positive-stranded single-stranded RNA (+ssRNA) that encodes two ORFs; ORF1a and ORF1b (*Kreuze, Savenkov & Valkonen, 2002*). The co-infection of SPFMV and SPCSV results in a synergistic reaction between these two viruses leading to the severe symptoms that are observed in *Sweet Potato Virus Disease* (SPVD), the most devastating viral disease of sweet potato (*Karyeija et al., 2000*; *Kreuze & Valkonen, 2017*). Another major virus found within the sweet potato production zones in east Africa is SPCFV (*Aritua et al., 2007*; *Aritua et al., 2009*). SPCFV has a single-stranded positive-sense RNA genome with filamentous particles of between 750 and 800 nm long and 12 nm wide (*Aritua et al., 2009*). The complete genome of SPCFV consists of 9,104 nt, and contains six putative ORFs (*Aritua et al., 2009*). Typical symptoms of SPCFV infection include fine chlorotic spots on the sweet potato leaves. Co-infection of SPCFV with SPCSV results in milder

symptoms compared to those observed in SPVD (*Tugume, Mukasa & Valkonen, 2016*). It is suspected that a whitefly vector is associated with the transmission of SPCFV (*Aritua et al., 2007*), however, vector transmission studies are yet to confirm this (*Aritua et al., 2007*; *Tugume, Mukasa & Valkonen, 2016*).

The agro-ecosystems in the western highlands of Kenya are characterised by a heterogeneous cropping system (*Tittonell et al., 2007*; *Wainaina et al., 2018*), which allow for virus movement between crops during the growing season. To date, there have been limited efforts to identify the diversity and phylogenomic relationships of plant viruses in this system. In addition, it is not known what the roles of recombination and selective pressure are in the evolution of these viruses. In this study, we used a high throughput sequencing approach to identify plant viruses within sweet potato, and sought to answer the question 'What is the phylogenomic relationship of sweet potato viruses present in the western highlands of Kenya, and what evolutionary states are they under?' Here, we report the first complete genomes of SPFMV, SPVC and SPCSV, and a partial SPCSV, from the western highlands of Kenya. In addition, we investigate the role of recombination and selective pressure across the complete genome in driving the evolution of these viruses.

These four viruses have previously been reported within east Africa, including Kenya (*Ateka et al., 2004*). However, detection was dependent on either immunoassay enzyme linked immunosorbent assays (ELISA) or polymerase chain reaction (PCR) amplification of the partial coat protein (CP) gene (*Ateka et al., 2004*; *Miano, LaBonte & Clark, 2008*; *Opiyo et al., 2010*). So far, there have been no complete genomes of these viruses reported from Kenya. Findings from this study will provide the basis for improving molecular diagnosis through better informed primer design and testing for a broader range of various virus strains within eastern Africa. In addition, the new genomes from this region will further contribute to the evolutionary analysis of this and other related sweet potato viruses.

## MATERIAL AND METHODS

### Field collection

Ethical approval to conduct this study was obtained from the University of Western Australia (RA/4/1/7475). In addition, permission to access all privately owned farms was obtained through signed consent forms by the head of each household. Sampling was carried out in the western highlands of Kenya over two cropping seasons (2015 and 2016) during the long season from April–August. Fieldwork activities were coordinated through the Cassava Diagnostics Project Kenyan node. We sampled 120 farms within this period as part of a larger field survey (*Wainaina et al., 2018*). A total of six viral symptomatic sweet potato samples were collected. The main viral symptoms observed on the leaves sampled in the fields were purple ringspots with leaf crinkling. For each symptomatic sample, two leaves were collected. One leaf of each sample was stored in silica gel, while the second leaf sample was stored using the paper press method (*Almakarem et al., 2012*). All samples were then transported to the BecA-ILRI hub laboratories in Nairobi, Kenya for virus testing.

## Nucleic acid extraction and PCR screening of viruses

From each individual leaf, RNA was extracted using the Zymo RNA miniprep kit (Zymo, Irvine, CA, USA) according to the manufacturers' specifications. Extractions were then lyophilised and shipped to the University of Western Australia for further processing.

Lyophilised RNA was subsequently reconstituted with nuclease free water. From an aliquot of the RNA, cDNA was prepared using Promega master mix (Promega, Madison, WI, USA) as described by the manufacturer. Subsequently, PCR was carried out using the Bioneer master mix (Bioneer, Daedeok, Republic of Korea) using two sets of primers; universal *Potyvirus* primers LegPotyF 5-GCWKCHATGATYGARGCHTGGG-3 and LegPotyR 5-AYYTGYTYMTCHCCATCCATC-3 (*Webster, 2008*) and for *Carlavirus* primers 5-GTTTTCCCAGTCACGAC-3 and 5-ATGCCXCTXAXXCCXCC-3 (*Chen, Chen & Adams, 2002*). *Bean common mosaic virus* was used as the positive control *Potyvirus* and a non-template control nuclease free water was used as the negative control.

## cDNA library preparation and RNA-Seq sequencing

A *cDNA* library was prepared from the a sweet potato sample that was positive after the initial PCR screening using Illumina Truseq stranded total RNA sample preparation kit with plant ribozero as described by the manufacturer (Illumina, San Diego, CA, USA). All libraries containing the correct insert size fragments and quantity were sent to Macrogen Korea for sequencing. Libraries were normalised based on concentration and then pooled before sequencing. Pair-end sequencing 2 × 150 bp was done on the rapid run mode using a single flow cell on the Illumina Hiseq 2500 Macrogen, Korea. However, four of the samples that were sent for sequencing failed at the quality control step of preparation and therefore did not proceed to sequencing. One of the remaining samples produced very low coverage, so we were unable to confidently undertake any analysis on that data. This left one single sample with good quality sequence for analysis.

## Assembly and mapping of RNA-Seq reads

Raw reads were trimmed and assembled using CLC Genomics Workbench (CLCGW ver 7.0.5) (Qiagen, Hilden, Germany). Trimmed reads were assembled using the following parameters: quality scores limit set to 0.01, the maximum number of ambiguities was set to two and read lengths less than 100 nt were discarded. Contigs were assembled using the de novo assembly function on CLCGW essentially as described in (*Kehoe et al., 2014a*; *Wainaina et al., 2018*). Reference-based mapping was then carried out using complete reference genomes retrieved from GenBank. Mapping parameters were set as follows: minimum overlap 10%, minimum overlap identity 80%, allow gaps 10% and fine-tuning iteration up to 10 times. The consensus contig from the mapping was aligned using MAFFT (*Katoh & Standley, 2016*) to the de novo contig of interest. The resulting alignments were manually inspected for ambiguities, which were corrected with reference to the original assembly or mapping. The ORF and annotation of the final sequences were done in Geneious 8.1.8 (Biomatters, Auckland, New Zealand). Sequences were referred to as nearly complete if the entire coding region was present, and complete if the entire genome including untranslated regions were present.

## Database retrieval of whole genome sequences

Whole genome sequences, of the four sweet potato viruses, were obtained from the National Centre of Biotechnology Information. The following sequences were obtained: SPFMV ($n = 25$), SPVC ($n = 20$), SPCFV ($n = 7$), and SPCSV ($n = 6$). Sequence alignment was carried out using MAFFT v7.017 (*Katoh & Standley, 2013*). The whole genome sequence alignments were deposited in zenodo DOI: 10.5281/zenodo.1254787.

## Detection of recombination breakpoints

Assessment of the recombination breakpoints of the nearly complete genomes from this study and those retrieved from GenBank was carried out using the seven programs within the RDP4 software (*Martin et al., 2015*). The programs used were: RDP (*Martin et al., 2005*), GENECONV (*Padidam, Sawyer & Fauquet, 1999*), Bootscan (*Martin et al., 2005*), MaxChi (*Smith, 1992*), Chimaera (*Posada & Crandall, 2001*), 3Seq (*Boni, Posada & Feldman, 2007*), and SiScan (*Gibbs, Armstrong & Gibbs, 2000*). A true recombination event was inferred if supported by at least four of the seven programs with a *P* value cut-off of 0.05 as described by previous studies (*Webster et al., 2007*; *Kehoe et al., 2014b*; *Maina et al., 2018b*).

## Bayesian phylogenetic analysis, coalescent species tree estimation using a coalescent framework and pairwise identity analyses

Bayesian inference was used to estimate the phylogenetic relationships for SPVC, SPFMV, SPCSV, and SPCFV. These analyses were carried out on the complete genomes and separately on individual genes. The most suitable evolutionary models were determined by jModelTest (*Darriba et al., 2012*). Bayesian analysis of the nearly complete genomes was carried out using Exabayes 1.4.1 (*Aberer, Kobert & Stamatakis, 2014*) while individual genes were analysed using MrBayes 3.2.2 (*Ronquist et al., 2012*). MrBayes was run for 50 million generations on four chains, with trees sampled every 1,000 generations using GTR +I+G as the evolutionary model. In each of the runs, the first 25% (2,500) of the sampled trees were discarded as burn-in. In the ExaBayes run, each gene segment was assigned an independent evolutionary model. ExaBayes was run was for 50 million generations on four chains. In each run, the first 25% of the sampled trees were discarded as burn-in. Convergence and mixing of the chains were evaluated using Tracer v1.6 (*Rambaut et al., 2014*) and trees visualised using Figtree (http://tree.bio.ed.ac.uk/software/figtree/).

Species tree estimation using the complete genome was carried out using Singular Value Decomposition (SVD) Quartets (*Chifman & Kubatko, 2014*) with a coalescent framework to estimate the species tree for SPFMV, SPCSV, SPVC, and SPCFV. The SVDQ analysis used all quartets with support of the species tree branches based on a bootstrap support of >50%. The species tree was visually compared to the gene trees from MrBayes and the complete genome tree from ExaBayes. Pairwise identities on the complete and partial sequences from Kenya, and from GenBank sequence were determined using Geneious 8.1.9 (Biomatters, Auckland, New Zealand).

## RESULTS

RNA-Seq on total plant RNA resulted in 12,667,976 reads which after trimming for quality came to 10,995,262 reads. De novo assembly produced 9,269 contigs from one sample (Table 1). Plant virus contigs were identified after BLASTn searches with lengths of between 8,427 and 16,157 nt, and had an average coverage of 1,339–11,890 times. Genome sequences with complete ORFs and complete untranslated regions (UTRs) were considered to be full genomes. However, genome sequences that lacked parts of the 5 and 3 UTR regions were considered to be near complete genomes. The final sequence was obtained from the consensus of de novo assembly and the mapped consensuses reads of 9,414–16,157 nt in length. The four sweet potato viruses obtained from this study are summarised in Table 1, and whole genome sequences retrieved from GenBank for analysis are summarised in Table S1. All viral sequences generated from this study were deposited in GenBank with the following accession numbers: SPVC (MH264531), SPCSV (RNA1 MH264532), SPCSV (RNA2, MH264533), SPCFV (MH264534), and SPFMV (MH264535).

### Analysis of recombination

Among the viral sequences from this study and those from GenBank, SPFMV was found to be recombinant at position 9, 9,964–10,482 nt within the CP region (Table 2). Moreover, the SPFMV sequences retrieved from GenBank were also found to be recombinant within the P1, Nla-Pro and CP gene regions (Table 2). The P1, Nla-Pro and CP genes were the hot spots of recombination.

### Bayesian Phylogenetic relationship, coalescent species tree estimation and percentage pairwise identity

Bayesian phylogenomic analysis among the sweet potato viruses was carried out across the whole genome in the case of SPVC, SPFMV, and SPCFV and within RNA1 and partial RNA2 in the case of SPCSV. Within SPCSV (RNA1 and RNA2) two well-supported clades were resolved, identified as clade I–II (Figs. 1 and 2). The Kenyan sequences clustered within clade II and were closely associated with two Uganda sequences and one sequence from China in both trees. Four well-supported clades identified as clades I–IV were resolved within the SPVC phylogenomic trees (Fig. 3). The Kenyan sequences clustered within clade II with sequences from Peru, Spain, and East Timor (Fig. 3). Three well-supported clades were resolved within the SPCFV phylogenomic tree, identified as clades I–III (Fig. 4). The Kenyan sequence clustered within clade III with two Ugandan sequences. Within the SPFMV phylogenomic tree comprising of both recombinant and non-recombinant sequences, two clades were resolved and identified as clades I–II (Fig. 5A). The Kenyan sequences were clustered in clade I. While phylogenomic analysis using *SPFMV* non-recombinant sequences resolved two well-supported clades that were associated with the main SPFMV strains, the russet crack (RC) clade I and the ordinary (O) clade II (Fig. 5B). The Kenyan sequence was excluded from this phylogenomic tree since it was recombinant. Moreover, phylogenetic analysis on the two genes where the recombination breakpoint was identified resolved two clades for the CP gene tree (Fig. 5C) and three clades for Nla-Pro gene tree (Fig. 5D). Within the CP gene tree,

**Table 1 De novo assembly and mapping of viral reads using CLC Genomic Workbench version 8.5.1 and Geneious 8.1.8.**

| Sample ID | Virus | Number of reads | Number of reads after trimming | Number of contigs produced | Reference sequence used for mapping | Length of consensus sequence from mapping (Geneious) | Number of reads mapped to reference sequence | Mean coverage (Geneious) | Contig positive for virus and length | Average coverage (CLCGW) | Number of reads mapped to contig of interest | % Similarity BLAST | Final sequence length |
|---|---|---|---|---|---|---|---|---|---|---|---|---|---|
| SRF 109a | SPFMV | 12,667,976 | 10,995,262 | 9,269 | FJ155666 | 11,424 | 890,045 | 11944.7 | 5(10,218) | 11,890 | 884,699 | 96 | 10,482 |
| | SPVC | | | | KU877879 | 11,410 | 466,349 | 6133.5 | 9(10,368) | 4,309 | 325,619 | 93;95 | 10,392 |
| | SPCFV | | | | KU720565 | 10,305 | 280,077 | 4383.5 | 19(8,427) | 5,430 | 335,367 | 97 | 9,414 |
| | SPCSV(RNA1)/RNA2 | | | | NC_004123 | 12,610 | 76,902 | 1169.4 | 85(16,157) | 1,339 | 164,959 | 99 | 16,157 |

**Note:**
The four sweet potato viruses identified were: *Sweet potato feathery mottle virus* (SPFMV), *Sweet potato virus C* (SPVC), *Sweet potato chlorotic fleck* (SPCFV) and *Sweet potato chlorotic stunt virus* (SPCSV).

**Table 2 Recombination signals across *Sweet potato feathery mottle virus* (SPFMV) using RDP4.**

| Recombination events | Recombinant sequence | Detected breakpoint | Parental sequence (major) | Parental sequence (minor) | Detected in RDP4 | Avr P-val |
|---|---|---|---|---|---|---|
| 1 | SPFMV_AB439206_Lab_Isolates SPFMV_MF572056.1_EastTimor | 5–1,004 | SPFMV_AB509454_Lab_Isolates | SPFMV_D86371_Lab_Isolates | **R**GBMC**S**3seq | 2.62 E-44 |
| 2 | SPFMV_KP115609_South_Korea | 22–948 | SPFMV_AB465608_South_Korea | SPFMV_MF572056.1_EastTimor | **R**GBMC**S**3seq | 1.41E-36 |
| 3 | SPFMV_MF185715.1_Brazil | 12–8,769 | SPFMV_MF572055.1_EastTimor | SPFMV_MF572054.1_Australia | RGBMC**S**3seq | 1.42 E-36 |
| 4 | SPFMV_KU511268_Spain | 7,062–7,946 | SPFMV_KP115608_South_Korea | SPFMV_AB509454_Lab_Isolates | RGBMC**S**3seq | 1.11 E-18 |
| 5 | SPFMV_KU511268_Spain | 51–7,061 | SPFMV_FJ155666_Peru | SPFMV_MF572054.1_Australia | RGBMC**S**3seq | 0.0042 |
| 6 | SPFMV_MF572055.1_EastTimor | 10,199–10,663 | SPFMV_MF572054.1_Australia | SPFMV_MF572046.1_Australia | RG**B**MC**S**3seq | 1.49 E-11 |
| 7 | SPFMV_MF572054.1_Australia | 10,218–10,663 | SPFMV_MF572049.1_Australia | SPFMV_SRF109a_Kenya | RGBMC**S**3seq | 1.30 E-09 |
| 8 | SPFMV_FJ155666_Peru | 1,642–7,476 | SPFMV_MF572054.1_Australia | SPFMV_AB465608_South_Korea | RGBMC**S**3seq | 1.53 E-09 |
| 9 | SPFMV_MF572056.1_EastTimor | 36–9,374 | SPFMV_MF572053.1_EastTimor | SPFMV_MF572052.1_Australia | **R**GBMC**S**3seq | 1.51 E-18 |
| 10 | SPFMV_FJ155666_Peru | 7,477–10,144 | SPFMV_SRF109a_Kenya | SPFMV_KY296450.1_China | R**G**BMC**S**3seq | 1.06 E-02 |
| 11 | SPFMV_SRF109a_Kenya | 9,696–10,216 | SPFMV_MF572050.1_Australia | SPFMV_KY296450.1_China | RGBMC**S3seq** | 1.31 E-07 |

**Notes:**

Table entries represent the recombinant sequences and the position of recombination within the complete genome. A recombination pattern was considered if supported by at least four of the seven RDP4 programs at a significance level of 0.05.

Recombinant programs in RDP4 that detected recombinant events across the whole genome of SPFMV **3**, 3seq; **B**, Bootscan; **C**, Chimera; **G**, Gencov; **R**, RDP; **M**, Maxchi; **S**, Siscan. Bold letters in the RDP column (detected in RDP4) indicate the program that detected the highest *P*-value.

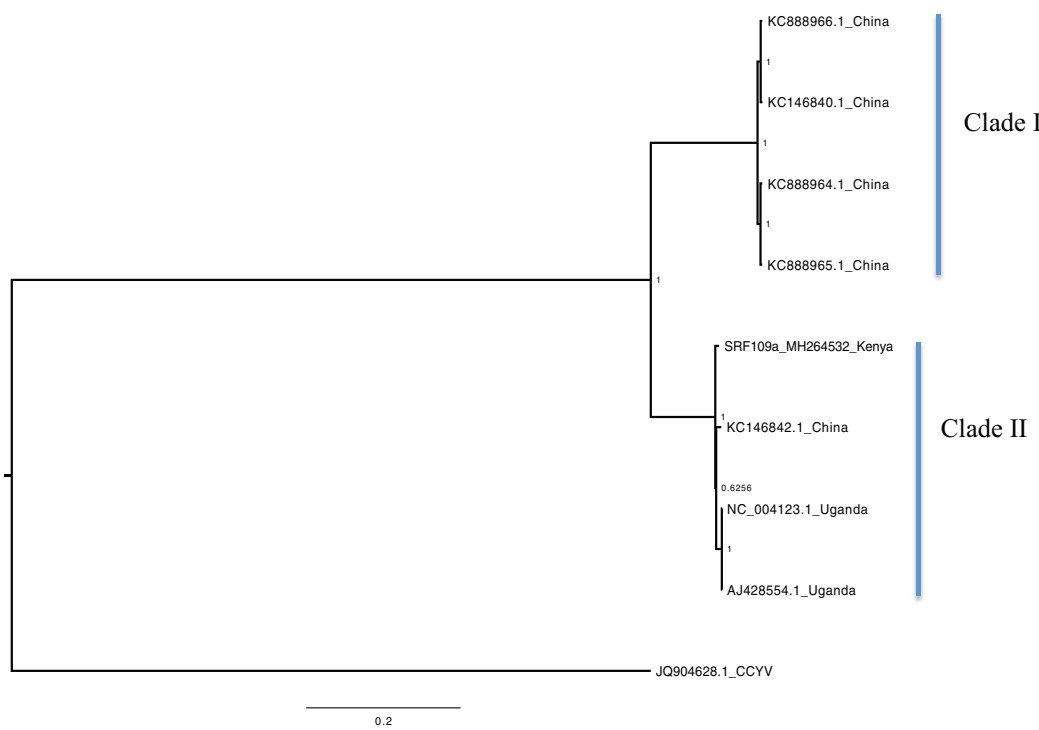

**Figure 1** Consensus of trees sampled in a Bayesian analysis of RNA 1 gene in *Sweet potato chlorotic stunt virus* (SPCSV) with *Cucurbit yellow stunting disorder virus* (CCYV) as the outgroup to root the tree. Scale bar on the phylogenetic tree represents the nucleotide substitution per every 100 sites. The nodes across each branch are labeled with posterior probability. Tip labels contain information of: virus name, GenBank accession number and/or field identification and country where sampling was conducted.                                   

recombinant sequence formed a distinct sub-clade identified as 1a within the larger clade I. While in Nla-Pro the recombinant sequence clustered in clade II (Fig. 5D). The CP gene is used as the primary target region for many virus diagnostic molecular markers, and this region tree resolved similar clades to both the concatenated genome tree and the coalescent species tree (Figs. S1–S4; Tables S2A–S2B) in SPCSV and SPCFV but not in SPVC and SPFMV (Tables S2A–S2B).

Percentage pairwise identities between the Kenya sequences and the GenBank sequences varied across the viruses within SPCSV RNA1 (83–99%), RNA2 (70–98%). The closest match to the Kenyan sequence was two Uganda sequences (AJ428554.1 and NC_004123.1) and a sequence from China (KC146843.1) with nucleotide identities of between 98.7% and 98.8%. Within SPVC nucleotide, identity match ranged between (91% and 98%). The closest match to the Kenyan sequence was a sequence from Spain (KU511269) with 93.3% percentage identity. Percentage nucleotide identity within the SPCFV ranged between 72% and 96%. The closest nucleotide identity matches to the Kenyan sequence were sequences from Uganda (NC_006550 and AY461421) with percentage identity of 96.5%. Percentage nucleotide identity within the SPFMV ranged between 87% and 98%. The closest nucleotide identity match to the Kenyan sequence was a sequence from China (KY296450).

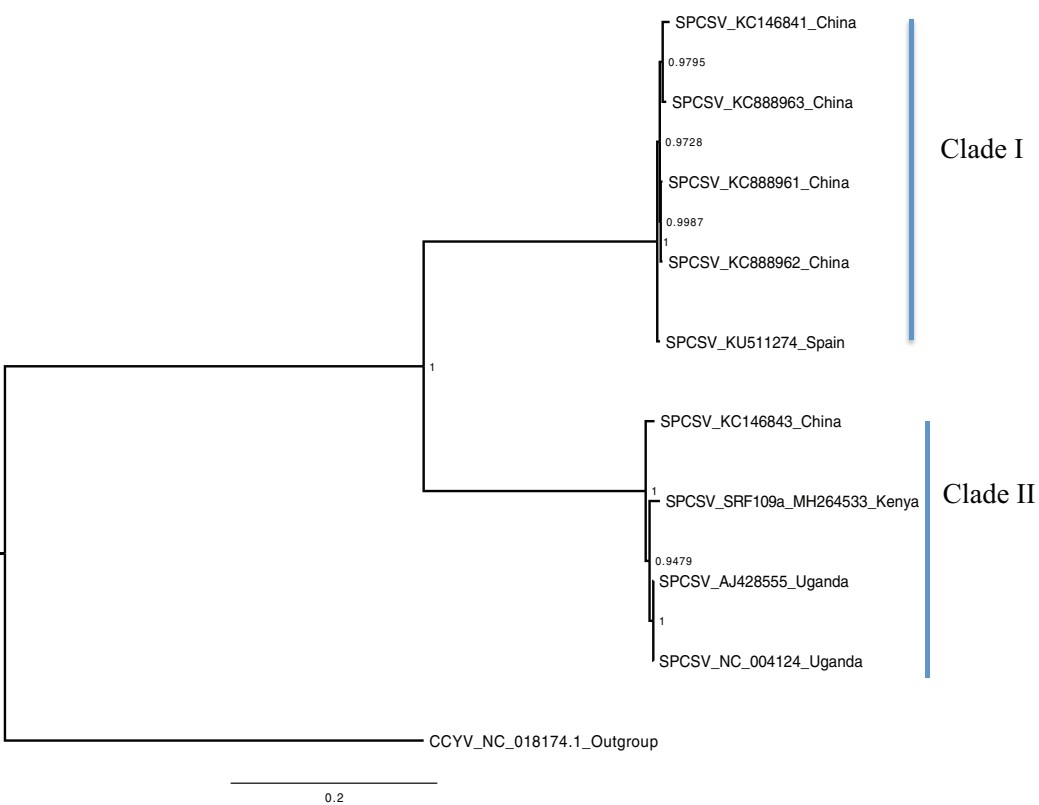

**Figure 2 Consensus of trees sampled in a Bayesian analysis of RNA 2 gene in *Sweet potato chlorotic stunt virus* (SPCSV).** *Cucurbit yellow stunting disorder virus* (CCYV) was used as the out-group to root the tree. The scale bar on the phylogenetic tree represents the nucleotide substitution per every 100 sites. The nodes across each branch are labeled with posterior probability. Tip labels contain information of: virus name, GenBank accession number and/or field identification and country where sampling was conducted.

## Selection pressure analysis across genes

Assessment of selective pressure based on the ratio of the average synonymous and non-synonymous ($d_N/d_S$) substitutions across the coding region of individual genes in each of the four viruses showed evidence of purifying selection (Figs. 6A–6D). However the rates of purifying selection ($d_N/d_S < 1$) were not homogeneous across genes. Genes that were under relative lower purifying selection were the P1 gene in both SPVC and SPFMV (Figs. 6A and 6D). On the other hand, triple block 3 and Nucleic acid binding virus genes in SPCFV (Fig. 6B) and the CP genes in all four viruses were under strong purifying selection with $d_N/d_S$ ratios of ~0.1 (Figs. 6A–6D). Purifying selection results in minimal changes to amino acids within the respective genes, which results in slow rates of evolution within these genes.

## DISCUSSION

One of the major limitations for sweet potato production, especially within smallholder agro-ecosystems in Kenya, is viral disease. Among these viral diseases is the SPVD attributed to the co-infection of SPFMV and SPCSV that act in synergy to exacerbate symptoms. In this study, we identified a mixed infection involving four viruses; SPFMV,

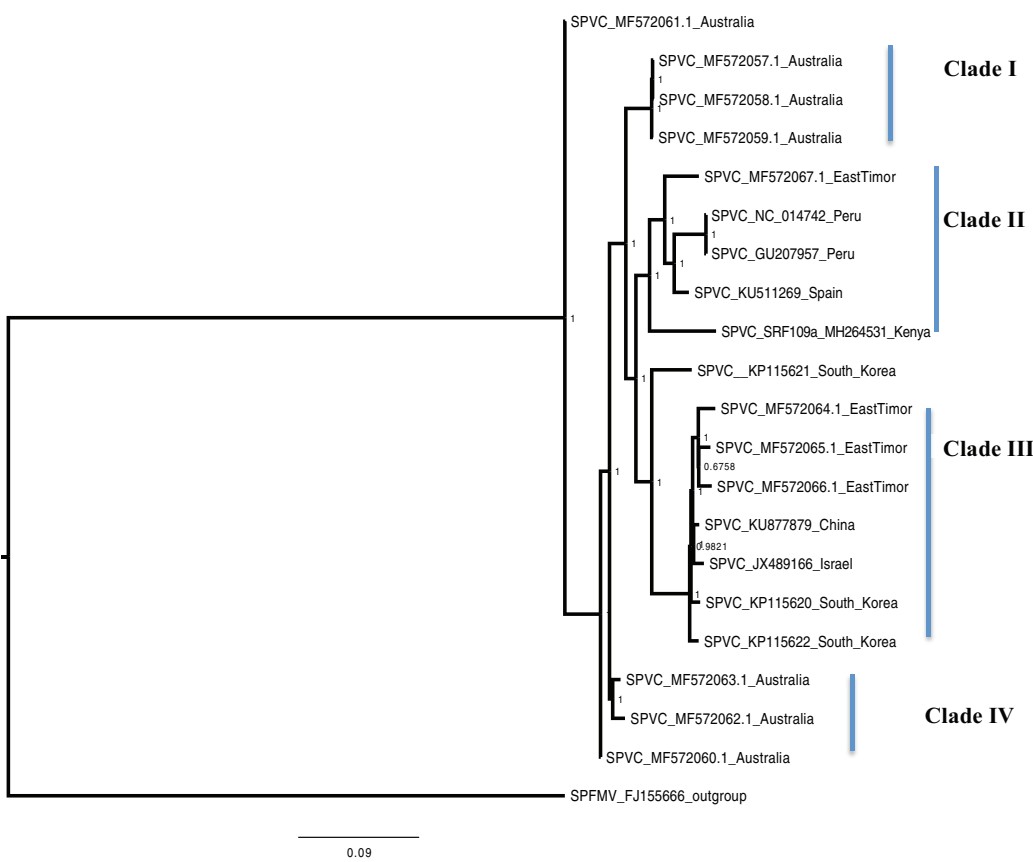

**Figure 3 Consensus of trees sampled in a Bayesian analysis of the whole genome of *Sweet Potato virus C* (SPVC) with *sweet potato feathery mottle virus* (*SPFMV*) used as the outgroup to root the tree.** Scale bar on each phylogenetic tree represents the nucleotide substitution per every 100 sites. The nodes across each branch are labeled with posterior probability. Tip labels contain information of: virus name, GenBank accession number and/or field identification and country where sampling was conducted.

SPCSV, SPVC, and SPCFV. We report the first complete genome of SPFMV, SPVC, SPCFV, and partial SPCSV from Kenya. The SPFMV and SPVC genomes are the first from sub-Saharan Africa. Moreover, we conducted phylogenomic relationship analysis of these genomes. In addition we identified recombination events and selective pressure as acting on the virus genomes and potential drivers for their evolution in Kenya and globally.

## High throughput RNA sequencing RNA-Seq on sweet potato

High throughput RNA sequencing (RNA-Seq) was used to identify the complete genome and partial genome of sweet potato viruses from a viral symptomatic sweet potato. We report the first complete genomes of SPVC (10,392 nt), SPFMV (10,482 nt), SPCFV (9,414 nt), and partial SPCSV (16,157 nt) (Table 1) from Kenya. Presence of the SPFMV and SPCSV are an indication of SPVD, being prevalent on the farm where sampling was done. SPVD remains one of the major diseases infecting sweet potato in eastern Africa. Previous reports of SPVD from the western highlands of Kenya and in the neighbouring regions of Uganda have been made (*Ateka et al., 2004*; *Opiyo et al., 2010*;

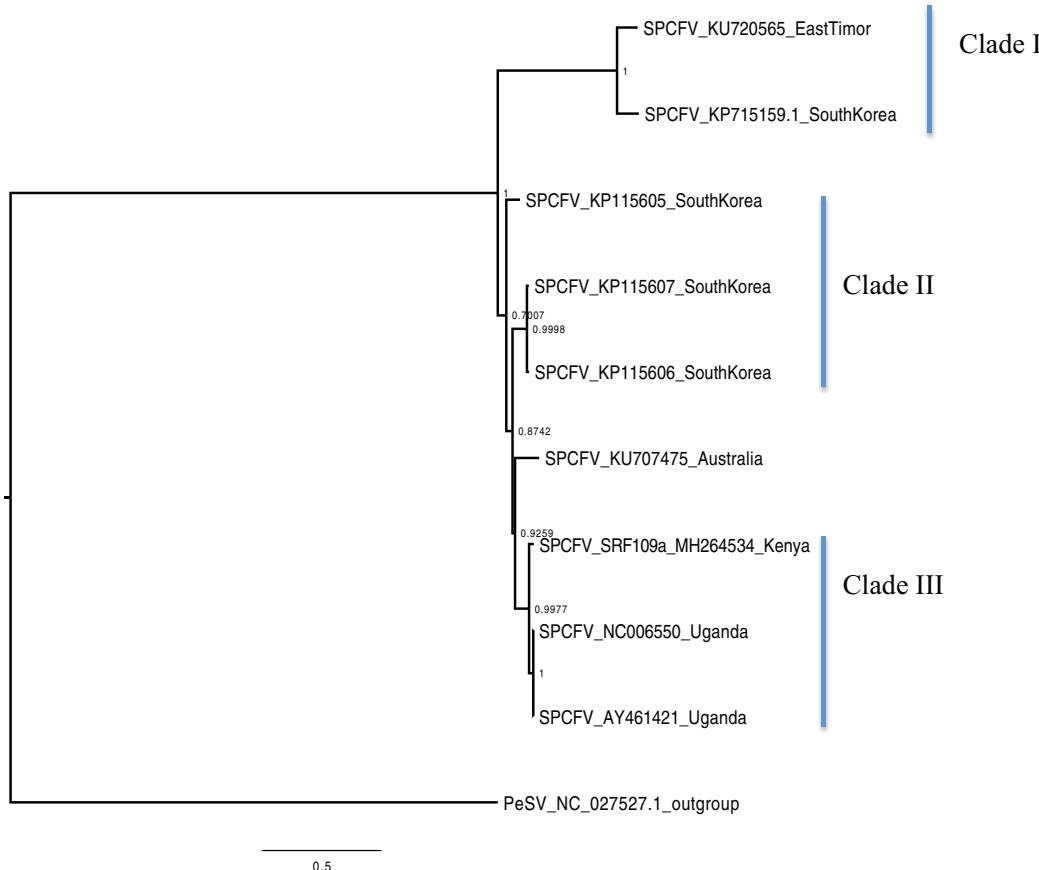

**Figure 4 Consensus of trees sampled in a Bayesian analysis of the whole genome of *Sweet potato chlorotic fleck* (SPCFV) virus with *Pea Streak virus* (PeSV) used as the outgroup to root the tree.** Scale bar on each phylogenetic tree represents the nucleotide substitution per every 100 sites. The nodes across each branch are labeled with posterior probability. Tip labels contain information of: virus name, GenBank accession number and/or field identification and country sampling where sampling was conducted.

*Tugume, Mukasa & Valkonen, 2016*). Prevalence of SPFMV were reported to be at 89% while SPCSV was 55% in Kenya using ELISA. In Uganda, the prevalence levels were between 1.3% for SPFMV and 5.4% in SPCSV based on next-generation sequencing. In this study, we build on these findings using a whole genome sequencing approach rather than single gene loci.

*Sweet potato feathery mottle virus* and SPVC belong to the family *Potyviridae,* and are spread by viruliferous aphids and through infected cuttings within sweet potato (*Ateka et al., 2004*). In addition, a *Carlavirus* SPCFV and partial *Crinivirus,* SPCSV were also identified (Table 1) with the primary mode of transmission being whitefly vectors coupled with infected cuttings (*Kreuze, Savenkov & Valkonen, 2002*; *Navas-Castillo, Lopez-Moya & Aranda, 2014*). Feeding of whitefly and aphids on the same plants results in the transmission of different viruses within that same host plant. This increases the chances of co-infection of multiple insect transmitted viruses. It is therefore likely that within the agro-ecosystems of western Kenya, there is heavy infestation of both aphids and whitefly

PeerJ ___________________________________________________________

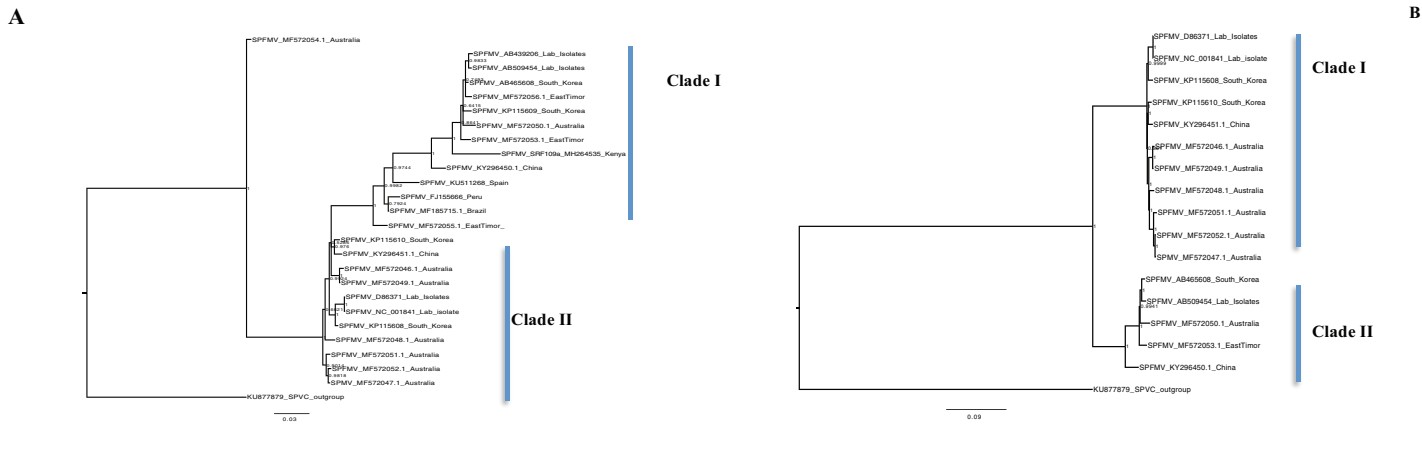

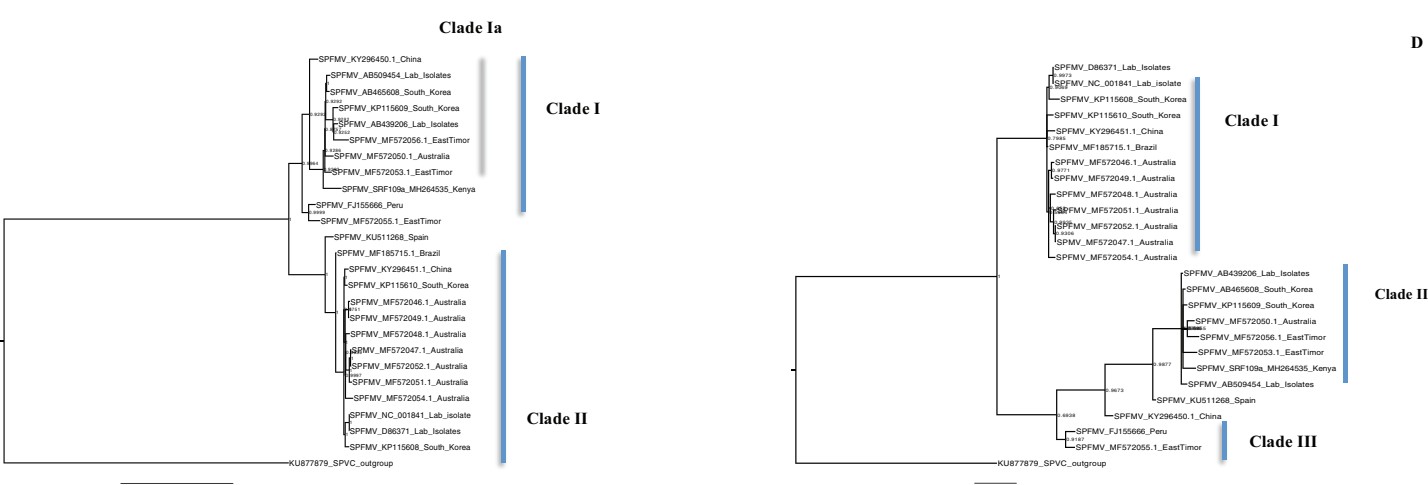

**Figure 5 Consensus of trees sampled in a Bayesian analysis of the whole genome phylogenetic tree.** (A) *Sweet potato feathery mottle virus* (SPFMV) with *Sweet potato virus C* (SPVC) as the out-group with both recombinant and non-recombinant sequences. (B) *Sweet potato feathery mottle virus* (SPFMV) with *Sweet potato virus C* (SPVC) as the out-group based on non-recombinant sequences. (C) Consensus of trees sampled in a Bayesian analysis of the coat protein gene of *Sweet potato feathery mottle virus* (SPFMV) with *Sweet potato virus C* (SPVC) as the outgroup using recombinant (clade Ia) and non-recombinant sequences. (D) Consensus of trees sampled in a Bayesian analysis of the Nla-Pro gene of *Sweet potato feathery mottle virus* (SPFMV) with clade II comprising of recombinant sequences that are evolving at different rates compared to non-recombinant sequences. Scale bar on each phylogenetic tree representative on the nucleotide substitution per every 100 sites. The nodes across each branch are labeled with posterior probability. Tip labels contain information of: virus name, GenBank accession number and/or field identification and country sampled.

vectors. Previous studies have reported aphid and whitefly-transmitted viruses in crops within the western region (*Legg et al., 2006*, *2014*; *Mangeni et al., 2014*; *Wainaina et al., 2018*) and the Lake Victoria region (*Tugume et al., 2010a*; *Adikini et al., 2015*; *Adikini et al., 2016*). Moreover, farming practices within smallholder farms, which include partial harvesting of mature sweet potato, are thought to help maintain the virus within the agro-ecosystem. The advantage of this practice is it allows for the crop to remain underground, where it stores well (*Loebenstein, 2010*), providing a sustainable food source for the

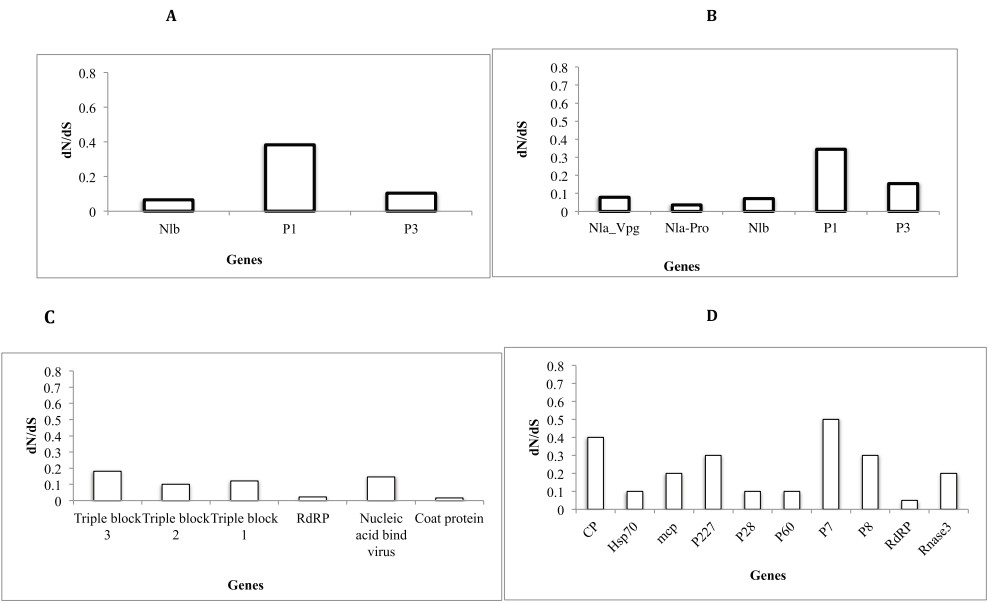

**Figure 6 Selection pressure within sites of the coding region of viral gene fragments determined by assessing the average synonymous and non-synonymous (dN/dS) using SLAC that were plotted against each gene segment.** (A) *Sweet potato virus C* (B) *Sweet potato feathery mottle virus* (SPFMV) (C) *Sweet potato chlorotic fleck virus* (SPCFV) (D) RNA1 and RNA 2 genes of *Sweet potato chlorotic stunt virus* (SPCSV). Genes with selection pressure of below 0.0 were not plotted.

farmers. However, a major drawback of these practices is that these sweet potato crops may act as potential viral reservoirs that then become a viral source that aids dissemination to non-infected host plants by insect vectors during the cropping season. This phenomenon results in the continuous circulation of viruses within the agro-ecosystems.

## Recombination in sweet potato viruses

Survival of plant viruses is dependent on their ability to be successfully transmitted to suitable host plants. Survival within the host plant is dependent on the ability of the virus to evade the host plant resistance system, while at the same time maintaining their genetic vigour to allow for replication. One approach that they utilise for their survival is recombination, which is a key driver of virus evolution and in addition to this, beneficial traits are acquired while deleterious ones are removed. Within the *Potyviridae,* recombination is highly prevalent (*Varsani et al., 2008*; *Elena, Fraile & García-Arenal, 2014*; *Ndunguru et al., 2015*; *Tugume, Mukasa & Valkonen, 2016*; *Wainaina et al., 2018*). Moreover, co-infection of multiple viruses, in particular within sweet potato, can result in well-adapted viruses and has been adversely reported in other countries (*Tugume et al., 2010a*; *Maina et al., 2018a*, *2018b*).

Analysis of recombination on both the new sequence and those retrieved from GenBank, identified 11 recombinant sequences in SPFMV (Table 2), which included the Kenyan sequence. The three other viruses identified (SPCV, SPVC, and SPCFV) from Kenya were not recombinant. The SPVC sequences from GenBank sequences were

recombinant but are well described and discussed elsewhere (*Maina et al., 2018b*). Within SPFMV, recombination was mainly found within P1, Nla-Pro and the CP region of the genome. These findings are consistent with previous SPFMV reports (*Maina et al., 2018a, 2018b*). The CP region is a hot spot of recombination mainly due to the selective pressure from the host immune system. As a strategy to evade the host immune system, the viral CP is constantly changing. On the other hand, the P1 gene is postulated to be the driver for diversity of the *Potyviruses*. This resulted in evolutionary branching of other members of the *Potyviruses* such as the *ipomovirus* and *tritimoviruses* (*Valli, López-Moya & García, 2007*). The main driver of recombination within the P1 region is postulated to be the interaction between the N-terminal region of P1 gene and the host plant (*Valli, López-Moya & García, 2007*). It is therefore common to have both intragenus and intergenus recombination within P1 thus facilitating better host adaption. Similarly, we postulate this could also be the primary reason for the recombination events within Nla-Pro. Nla-Pro is associated with the proteolytic activities within members of the family *Potyviridae*. In addition, it regulates the potyviral proteins at different stages of infection thus ensuring successful viral colonisation (*Ivanov et al., 2014*).

## Phylogenomic relationship between sweet potato viruses

Phylogenetic analyses were carried out between the complete genomes from Kenya and reference GenBank sequences (Figs. 1–5). In both, SPCSV RNA1 and RNA2 (Figs. 1 and 2) and SPCFV (Fig. 4) Kenya and Uganda sequences clustered together in well-supported clades. The percentage nucleotide similarity was over 96% compared to Uganda sequences. We suggest the clustering of Uganda and Kenya sequences could be due to movement of infected plant cuttings across the borders of Kenya and Uganda. Communities living in this region have a shared kinship that transcends the geopolitical borders and often there is exchange of vegetative planting material. Moreover, there is inadequate phytosanitary screening across the borders for plant cuttings. Previous studies have reported both virus and vector movement through plant cuttings along these border regions (*Legg et al., 2011*). In addition, this mode of virus spread has also been reported in other vegetatively propagated crops such as cassava (*Legg et al., 2014*; *Alicai et al., 2016*).

*Sweet potato virus C* sequences from this study clustered with the South-American Peru, Spanish, and one East Timor sequence in a single well-supported clade (Clade II) (Fig. 3) with the closest similarity a sequence from Spain (KU511269) with 93% nucleotide identity. SPVC is likely to have been introduced into the eastern Africa regions through trade, and the British colonialists and missionaries, with the introduction of sweet potato into eastern Africa. The Portuguese traders transported sweet potato from South America to Africa through Mozambique and Angola around the 15th century (*Loebenstein, 2010*). The British colonialists subsequently followed them in 1662. We hypothesize SPVC may then may have jumped into the native vegetation, and has been maintained within the agro-ecosystem since that time. More recently, international trade between Kenya, Europe and parts of South America, is a possible route for the continued introduction of SPVC into the western highlands of Kenya. More SPVC genomes

sequenced across more geographical regions will in future provide an opportunity to better understand the evolutionary dynamics of this virus.

The phylogenomic relationship of SPFMV sequences is possibly distorted due to the presence of recombinant SPFMV sequences (Table 2). Recombination has been implicated in misrepresenting the true phylogenetic relationship of viruses (*Schierup & Hein, 2000*; *Posada & Crandall, 2002*; *Varsani et al., 2008*). In this study, SPFMV sequences both from this study and GenBank were found to be recombinant (Table 2). Recombinant sequences formed a distinct clade on both the CP and Nla-Pro gene trees (Figs. 5C–5D) and whole genome tree (Figs. 5A–5B). A significant feature of recombination on the phylogenetic tree is the splitting of sequences into recombinant versus non-recombinant clades, which was observed (Figs. 5A, 5C, and 5D). Thus any inference in the clustering of SPFMV sequences, in particular, with recombinant sequences present is likely to be inaccurate. The SPFMV phylogenomic tree with non-recombinant sequences resolved two clades associated with two of the three main phylogroups present in SPFMV associated with the SPFMV strains RC and O (*Kreuze et al., 2000*; *Maina et al., 2018a*) (Fig. 5B).

Single gene loci are used in routine molecular diagnostics and subsequent analysis of the phylogenetic relationship of viruses. A majority of the gene trees across all four viruses were discordant to the concatenated genome tree except within the CP gene which is the primary diagnostic marker (*Colinet et al., 1995*). However there was concordance between the number of clades resolved from the concatenated whole genome tree, the coalescent species tree, and the CP gene trees in SPCSV (RNA1 and RNA2) and SPCFV (Table S2B) however, this was not the case in SPFMV and SPVC (Table S2B). The discordance between the gene trees and the species trees could be attributed to; incomplete lineage sorting (ILS), gene gain and loss, horizontal gene transfer (HGT) and gene duplication (*Maddison, 1997*). It is probable that some of these factors could be the difference between the gene and species trees. These findings support the use of the CP as an ideal diagnostic marker for molecular diagnostics within SPCSV and SPCFV. Our findings are comparable to previous virus whole-genome studies (*Wainaina et al., 2018*). However, they also differ with other viruses within the *Potyviridae,* for example within *ipomoviruses* such as the cassava brown streak virus and Uganda cassava brown streak virus (*Alicai et al., 2016*). A probable cause of these differences could be the divergence of the *Ipomoviruses* from other members of the family *Potyviridae*. Therefore, it is necessary to evaluate all gene trees against the coalescent species tree and concatenated genome tree of individual viruses. This will aid in determining which of the genes reflects the true phylogenetic relationship of the virus based on the sequences. This approach is more stringent, and provides a robust analysis to choose a suitable gene region from which to create new diagnostic tools. This is imperative for the control and management of plant viral infections.

## Selection pressure analysis between genes of the sweet potato viruses

Selective pressure across genes of RNA viruses varies across viral families and genes (*Duffy, Shackelton & Holmes, 2008*). Though RNA viruses undergo rapid evolutionary

rates, this is dictated by several factors such as viral populations, inter versus intra-host variation, and population sizes (*Duffy, Shackelton & Holmes, 2008*). Across all the viral sequences (Figs. 6A–6D) the CP genes were under strong purifying selection $d_N/d_S$ ~ 0.1. This strong purifying selection is evident in a majority of vector-transmitted viruses, due to the fitness trade-off phenomena (*Chare & Holmes, 2004*). The fitness trade-off states that due to the limited number of insect vectors and specificity between the insect vectors and viruses that transmit RNA viruses, the evolution of the RNA viruses is constrained by their insect vectors (*Power, 2000*; *Chare & Holmes, 2004*). While deleterious mutations occurring within the RNA viruses could potentially affect their transmission, they are removed through purifying selection (*Chare & Holmes, 2004*). Purifying selection is more pronounced within the CP as previously reported (*Chare & Holmes, 2004*; *Alicai et al., 2016*; *Wainaina et al., 2018*). This further supports the hypothesis of the fitness trade-off phenomena in particular within plant RNA viruses with insect vectors.

On the other hand, within SPFMV and SPVC from the family *Potyviridae* we identified the P1 gene region to be under the least selection pressure (Figs. 6A and 6B). This indicates that though purifying selection was evident within the P1 gene, it was to a lesser extent compared to the CP gene. P1 is associated with viral adaptation of the host plant (*Shi et al., 2007*, *Salvador et al., 2008*; *Tugume et al., 2010b*), and it interferes with the host plant RNA induced silencing complex (*Tugume et al., 2010b*). This helps to ensure that viruses can evade the host immune response. This increases the chances for the virus to establish itself and survive within the host plant. Mutations that may facilitate survival of the virus are therefore tolerated within the P1 region. Overall, all genes within the SPCFV were under strong purifying selection.

## CONCLUSION

We used high throughput sequencing on viral symptomatic sweet potato plants collected within the western highlands of Kenya. We identified co-infection of SPCSV, SFMV, SPVC, and SPCFV and obtained the first complete genome of these viruses from Kenya. Moreover, percentage nucleotide identity in SPCSV and SCFV sequences from Kenya were closely matched to sequences from Uganda with nucleotide similarity of above 96%. Inadequate phytosanitary measures and a porous border between Kenya and Uganda are likely factors that contribute to and further exacerbate the problem. The SPVC whole genome from this study clustered with sequences from South America. We postulate that SPVC may have been introduced into eastern Africa from the initial sweet potato cultivars from South America. SPVC was subsequently maintained within native vegetation and by vegetative propagation after the initial viral jump. Evolutionary insights based on recombination events and selective pressure analysis revealed the following; within all four viruses, only SPFMV sequences were found to be recombinant. This was especially within the P1, Nla-Pro and CP genes. Recombinant SPFMV sequences formed a distinct clade on both the whole genome tree and the gene trees, particularly within the Nla-Pro and CP genes. Conversely, selection pressure analysis across the genes varied across all four viruses. The CP gene was under strong purifying selection in all viruses, while the P1 gene

in SPFMV and SPVC showed weak positive selection. Our findings provide a snap shot of viruses present within sweet potato and a more extensive study within the western highlands of Kenya would most likely reveal more extensive viral infections within this region.

Future studies should be conducted within the Lake Victoria region and the western highlands of Kenya, to identify all possible sweet potato viruses and potential viral reservoirs within this region. A combination of both sequencing using the Oxford nanopore sequencing technology (*Boykin et al., 2018*), ELISA, and Loop mediated isothermal amplification, may provide faster and more cost effective approaches for the detection of multiple viruses within symptomatic sweet potato. This is especially important within east Africa where multiple viral infections are prevalent in most vegetatively propagated crops. Moreover, the availability of more viral sequences within this region will allow for further viral evolution studies to be conducted. This information will be crucial in determining when the viruses undergo changes and what the drivers of these changes are within the agro-ecosystems.

### Funding
James M. Wainaina is supported by an Australian Award Scholarships from the Department of Foreign Affairs and Trade (DFAT), and this work is part of his PhD research. Pawsey Supercomputing Centre provided supercomputer resources for data analysis with funding from the Australian Government and the Government of Western Australia. Laboratory and sequencing cost were paid for through a Rising star grant from the Faculty of Science University of Western Australia to Laura M. Boykin. The funders had no role in study design, data collection and analysis, decision to publish, or preparation of the manuscript.

### Grant Disclosures
The following grant information was disclosed by the authors:
Australian Award Scholarships from the Department of Foreign Affairs and Trade (DFAT).
Australian Government and the Government of Western Australia.
Faculty of Science University of Western Australia.

### Competing Interests
Laura M. Boykin is an Academic Editor for PeerJ.

### Author Contributions
- James M. Wainaina conceived and designed the experiments, performed the experiments, analyzed the data, contributed reagents/materials/analysis tools, prepared figures and/or tables, authored or reviewed drafts of the paper, approved the final draft.
- Elijah Ateka performed the experiments, contributed reagents/materials/analysis tools, authored or reviewed drafts of the paper, approved the final draft.

- Timothy Makori performed the experiments, contributed reagents/materials/analysis tools, authored or reviewed drafts of the paper, approved the final draft.
- Monica A. Kehoe conceived and designed the experiments, performed the experiments, analyzed the data, contributed reagents/materials/analysis tools, prepared figures and/or tables, authored or reviewed drafts of the paper, approved the final draft.
- Laura M. Boykin conceived and designed the experiments, performed the experiments, analyzed the data, contributed reagents/materials/analysis tools, prepared figures and/or tables, authored or reviewed drafts of the paper, approved the final draft.

## Field Study Permissions

The following information was supplied relating to field study approvals (i.e., approving body and any reference numbers):

Ethical approval to conduct this study was obtained from the University of Western Australia (RA/4/1/7475). In addition, permission to access all privately owned farms was obtained through signed consent forms by the head of each household. Sweet potato samples were collected as part of a larger field survey in the western highlands of Kenya over two cropping seasons (2015 and 2016) during the long season (*Wainaina et al., 2018*).

## DNA Deposition

The following information was supplied regarding the deposition of DNA sequences:

GenBank with the accession numbers: SPVC (MH264531), SPCSV (RNA1 MH264532), SPCSV (RNA2, MH264533), SPCFV (MH264534), and SPFMV (MH264535).

## Supplemental Information

Supplemental information for this article can be found online at http://dx.doi.org/10.7717/peerj.5254#supplemental-information.

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
