# Peer review of "Phylogenomic relationship and evolutionary insights of sweet potato viruses from the western highlands of Kenya"

_PeerJ, doi:10.7717/peerj.5254_

## Round 0.1 · original submission · Minor Revisions

I will be glad to receive a point by-point answer to all the reviewers suggestions. Please pay particular attention to those related to format. Also attend the request for availability of sequences and alignments. I agree that your study is relevant enough to merit publication in PeerJ. I expect your modified manuscript and then I will give my decision.

Reviewer 1 ·

Basic reporting

1. Genome assemblies have been deposited in Genbank, which is commendable, but they do not appear to have been made public yet. The short-read data must also be archived, particularly given that non-free closed-source software was used for assembly. The raw reads presumably contain quite a bit of information about virus sequence variation in the plant sampled (beyond the consensus sequences assembled).

2. Providing the multiple alignments constructed (e.g. in some repository such as Zenodo or figshare) would greatly increase the utility of this analysis. Presumably the alignments were carefully inspected and adjusted, but there is no evidence of this in the text.

3. Tree sizes, font, text size, and panel labels are wildly inconsistent (Figures 1 to 5, supplemental figures 1 to 5). Multiple clade-label lines are ambiguous.

4. No interpretation is provided in the "Analysis of recombination" and "Selection pressure analysis across genes" results sections. Consider moving text from the current discussion into those sections. Breaking up the discussion with section headings and folding the current conclusion section into it could further help balance out the manuscript.

5. Manuscript needs more careful proof-reading.

Experimental design

The sampling method used for this observational study is briefly sketched at the start of the methods section but lack of detail prevents evaluation of the scope of inference. How many samples were collected? How many were judged potyvirus-positive? Just one, as implied by line 162? (A grammatical error renders that sentence ambiguous: "the a sweet potato sample that was positive".) What controls were used for RT-PCR? I see that the authors cite their other preprint for sampling methodology, but that manuscript appears to suffer from a similar lack of detail. The single plant used for RNA-sequencing is described as symptomatic, but no indication is given whether symptoms were typical of infection with the viruses emphasized in the text. The rationale for how sequence data from this single plant can suffice for generalizing about the western highlands of Kenya (per the question stated in lines 122 to 124) is not clear.

Genomic analysis in the absence of epidemiological information is fine, but the current manuscript purports to range far beyond that.

Validity of the findings

1. The relevance of these results for assessing virus spread by vegetative propagation seem unclear, for the reasons stated above. Lines 474 to 477 appear overstated; compare to lines 389–392.

2. Mixed infection with four viruses from distinct species sounds like a lot, but is it truly surprising? It seems essential to better contextualize this observation and avoid implying that sequencing is necessary or cost-effective for assaying mixed infection.

3. I find the "four out of seven programs agree" criterion for calling recombination strange and the averaging of p-values even more puzzling. Consider adding some justification for these choices.

Additional comments

1. 3D barcharts (Figure 6) are not a perceptually effective type of data display.

2. Per the ICTV "How to write a virus name" document, capitalize the first letter of the first word in the name. Distinguishing the species from the viruses discussed in a few places might clarify meaning.

3. Consider removing "and evolutionary insights" from the title.

Reviewer 2 ·

Basic reporting

Specific comments

Abstract: Line 59 to 61: The statement “Our findings demonstrate the need for clean planting materials as the first line of control….” is not only misleading but also off-of track. The study did not explore anything about clean seed systems, or virus control. Much as a field survey was done, but the paper has predominantly reported computational evidence and for evolutionary insights of sweetpotato viruses. The recommendation should therefore be inline with what the study discovered. There is nothing about seed systems that the research discovered.

Main text:
The presentation of results and writing style is excellent. However referencing style need to be changed, especially when quoting three or more authors. For example “Tugume, Mukasa and Valkonen, 2016”, this should have been written Tugume et al., 2016. There are many areas where authors need to change this kind of referencing style.

Virus names should be italicized through out. In some places this was forgotten, for example Lines 98, 333.

Line 145: The heading omitted ‘of’ between screening and virus.

After presenting the virus names in full for the first time in the text, the virus abbreviations should be used through out and not repeating the full virus names as has been done many times in the text (Lines, 105, 189-191, 206, 207, 309-311).

Lines 280 – 283: “The discordance between the gene trees and the species trees could be attributed to……This statement is a misplacement. It should have been under discussion section.

Figures: The molecular evolutionary trees (Figs 1 to 5) should be informative. Explain what the scale mean. Is it substitutions/site?. Also the node values…..what are they? bootstrap values? explain them in legend.

Experimental design

The study was executed well with clear design. However, the authors need to explain number of field samples collected, number of leaf samples used in the lab analysis. This information is key in research design.

Validity of the findings

Findings could be validated, but details on virus survey as I indicated above (under experimental design) should be presented.

Additional comments

General comments
This manuscript report the findings of a virus study conducted in Kenya. While this paper clearly shows a huge effort in terms of computational analysis, the information on field survey is somehow limited. The paper should present details of the field survey especially sampling strategy, number of samples collected, number of leaf samples used for laboratory extractions, and number of isolates used for sequencing. This information is key in validating the study and providing insights into the field management and epidemic spread of the viruses, which the center pivot of phytopathological research and development.
This study adds value to evolution of sweetpotato viruses in Kenya, and application of the results are substantial in computational evolutionary studies. I commend the authors for their good computational analysis. In addition, the manuscript is clearly written in professional, unambiguous language. Otherwise, I recommend the paper for publication with few revisions as highlighted in the specific comments section.

Annotated reviews are not available for download in order to protect the identity of reviewers who chose to remain anonymous.

---

## Round 0.2 · accepted · Accept

Dear Laura,

Thank you for communicating your work to Peer. Your study adds on to the current knowledge of the evolution of sweetpotato viruses in Kenya. I can see you have responded properly the reviewers comments and I am happy to accept you manuscript for publication.

#